# Normative Values for Sternoclavicular Joint and Clavicle Anatomy Based on MR Imaging: A Comprehensive Analysis of 1591 Healthy Participants

**DOI:** 10.3390/jcm13123598

**Published:** 2024-06-19

**Authors:** Theo Morgan Languth, Anne Prietzel, Robin Bülow, Till Ittermann, René Laqua, Lyubomir Haralambiev, Axel Ekkernkamp, Mustafa Sinan Bakir

**Affiliations:** 1Center for Orthopaedics, Trauma Surgery and Rehabilitation Medicine, University Medicine Greifswald, Ferdinand-Sauerbruch-Straße, 17475 Greifswald, Germany; s-thlang@uni-greifswald.de (T.M.L.); anne.prietzel@med.uni-greifswald.de (A.P.); lyubomir.haralambiev@med.uni-greifswald.de (L.H.); ekkernkamp@ukb.de (A.E.); 2Department of Anesthesiology, University Medical Center of the Johannes Gutenberg University Mainz, Langenbeckstr. 1, 55131 Mainz, Germany; 3Institute of Diagnostic Radiology and Neuroradiology, University Medicine Greifswald, Ferdinand-Sauerbruch-Straße, 17475 Greifswald, Germany; robin.buelow@med.uni-greifswald.de; 4Institute for Community Medicine, University Medicine Greifswald, Ferdinand-Sauerbruch-Straße, 17475 Greifswald, Germany; till.ittermann@uni-greifswald.de; 5Institute of Diagnostic Radiology, Städtisches Krankenhaus Kiel, Chemnitzstraße 33, 24116 Kiel, Germany; rene.laqua@krankenhaus-kiel.de; 6Department of Trauma Surgery and Orthopedics, BG Hospital Unfallkrankenhaus Berlin gGmbH, Warener Straße 7, 12683 Berlin, Germany

**Keywords:** clavicular anatomy, sternoclavicular joint, reference values, MRI diagnostics, SHIP

## Abstract

**Background:** The clavicle remains one of the most fractured bones in the human body, despite the fact that little is known about the MR imaging of it and the adjacent sternoclavicular joint. This study aims to establish standardized values for the diameters of the clavicle as well as the angles of the sternoclavicular joint using whole-body MRI scans of a large and healthy population and to examine further possible correlations between diameters and angles and influencing factors like BMI, weight, height, sex, and age. **Methods:** This study reviewed whole-body MRI scans from the Study of Health in Pomerania (SHIP), a German population-based cross-sectional study in Mecklenburg–Western Pomerania. Descriptive statistics, as well as median-based regression models, were used to evaluate the results. **Results:** We could establish reference values based on a shoulder-healthy population for each clavicle parameter. Substantial differences were found for sex. Small impacts were found for height, weight, and BMI. Less to no impact was found for age. **Conclusions:** This study provides valuable reference values for clavicle and sternoclavicular joint-related parameters and shows the effects of epidemiological features, laying the groundwork for future studies. Further research is mandatory to determine the clinical implications of these findings.

## 1. Background and Introduction

Advancements in magnetic resonance imaging (MRI) diagnostics have been noteworthy, with increasing recognition of its efficacy [1,2]. Offering superior detection capabilities for soft tissue damage without subjecting the patient to radiation, MRI has become an invaluable tool [3,4,5,6]. Since standard practice for assessing bony injuries or fractures involves X-ray examinations, commonly referenced tables are predominantly derived from X-ray data [7,8]. These reference data are challenging to use with increasing CT scans and MR imaging, where reference tables are missing. The rising utilization of MRI in shoulder girdle pathologies and its use for clavicular joint injuries or fractures with suspected neurovascular complications raises the question of whether there are potential disparities between reference tables generated from MRI and X-ray data [9,10,11]. Although cadaveric dissection remains an alternative, it demands substantially more effort, often with a smaller subject pool, and lacks comparability to real diagnostic scenarios [12,13,14,15]. 

This study aims to juxtapose the normative values obtained from X-ray diagnostic assessments of the sternoclavicular joint (SCJ) with those from MR diagnostics. Notably, bone elements exhibit variations in their MRI comparability. Leveraging data from the extensive Study of Health in Pomerania (SHIP), this research incorporates various epidemiological hypotheses, analyzing the influence of sex, age, weight, height, and BMI on clavicle and SCJ anatomy. This potentially contributes to the development of a novel reference table and provides new data for the interpretation of symmetry in anatomy along with the effects of these distinct traits. Despite the limited scientific literature on SCJ and clavicular anatomy, even fewer studies employ MRI to evaluate diverse lengths and angles, providing a lack of information. Surgeons undertaking the task of restoring anatomical conditions, especially in the context of emerging fracture treatment techniques for the human clavicle, one of the most common bony injuries, must possess a comprehensive understanding of normal anatomical averages [16,17]. Current research trends hint at the superiority of operative over nonoperative treatment for clavicle fractures concerning particular outcome parameters, emphasizing the urgency for refining radiological analyses [18,19,20]. Such improvements are fundamental prerequisites for informed decision-making and are crucial in adapting and advancing anatomically contoured plate osteosynthesis techniques. Furthermore, improved knowledge of the range of physiological MRI findings could contribute to diagnosing patients with pain in the region of the SCJ and the clavicle. Reference values are necessary to differentiate and separate normal variations from pathological ones, differentiating the origins of pain [21,22]. As a clinical example of MRI advantage, special lesions like bone marrow infiltration are only seen in an MRI but not in a CT scan, emphasizing their value [7]. Thus, the pivotal questions persist: if the structures resemble MRI imaging, if there are epidemiological features with significant implications, and if the measurements deviate from X-ray imaging.

## 2. Material and Methods

### 2.1. Design and Sample

The ongoing population-based project SHIP encompasses two independent cohorts, namely SHIP and SHIP-Trend [23]. Rigorous participant selection from official resident registry office files ensures representation mirroring the average population of Mecklenburg-Vorpommern in Germany. Key demographic variables such as sex, age, height, weight, and city of residence were meticulously characterized [23,24].

In the baseline assessment (SHIP-0) executed between 1997 and 2001, n = 4308 adults participated, with subsequent follow-up examinations in SHIP-1 (2002–2006, n = 3300) and SHIP-2 (2008–2012, n = 2333). Additionally, the SHIP-Trend cohort, initiated in 2008 with 4420 participants, underwent similar assessments [23,24]. Combining SHIP-2 and SHIP-Trend, n = 3371 out of 6753 participants underwent MRI examinations, with exclusions for claustrophobia, metal implants, or personal reasons. This study utilized MRIs from a final cohort of n = 2436 individuals.

For the reference table, patients exhibiting edema, effusion, perimeter extensions, glenoid retroversion, nearby fractures, or shoulder pain intensity >3 (on a scale of 1 to 10) were excluded in order to define a shoulder-healthy population. 

The Magnetom Avanto 1.5 Tesla by Siemens Medical Systems, Germany, was used for MRI scans and evaluated independently by two radiologists using standardized examination sheets [25]. 

MRI images from SHIP-TREND and SHIP-2 were analyzed using a plug-in designed for HOROS™ (Version 3.3.6; Nimble Co. LLC, d/b/a Purview, Annapolis, MD, USA). This plug-in facilitated distance and angle measurements between various anatomical points. Transversal layers were initially created, supplemented by an additional 3D frontal layer generated by the study authors using HORUS PACS. The plug-in was used in a similar way before [26].

### 2.2. Measurement Parameters

The retrospective cross-sectional study focused on eight key parameters, each elucidating different aspects of the sternoclavicular joint and clavicle anatomy.

Sternal angle: The sternal angle alpha was measured by the plug-in after the two tangents were drawn. Tangent 1 aligns at the sternal articular surface of the sternoclavicular joint, and tangent 2 is at the cranial end of the manubrium (Figure 1).

2.Clavicular angle: The clavicular angle was measured by the plug-in after two tangents were drawn. One aligns at the articular surface of the clavicular end of the sternoclavicular joint, and the other one at the cranial border of the medial end of the clavicle (Figure 2).

3.Clavicular–sternal angle: The lengthwise axis of the clavicle is determined by a line, which is drawn from the middle of the lateral clavicle to the middle of the medial clavicle. The vertical axis of the sternum is drawn from the middle of the cranial end of the sternum to the xiphoid process of the sternum. Those two axes created the obtuse angle gamma (Figure 3).

Different parameters have been analysed to describe the clavicle and its characteristic diameters: the maximum medial and lateral diameter and the minimal diameter in the midshaft (Figure 4). 

4.Max. medial diameter: The distance was measured by drawing a line between the most dorsal and the most ventral points of the medial end of the clavicle (Figure 5).

5.Min. diameter: The distance was measured by drawing a line between the least ventral and least dorsal points of the clavicle (Figure 6).

6.Max. lateral diameter: The distance was measured by drawing a line between the most ventral and most dorsal points of the lateral clavicula (Figure 7).

The overall clavicular maximal diameter (MAXCL) was determined by comparing the maximum lateral and medial diameters.

### 2.3. Statistics

To assess reliability, n = 106 images were measured twice by two readers, demonstrating interreader variability within acceptable limits. Bland and Altman plots and interclass correlation coefficients were applied to evaluate intra- and interreader variabilities (Figure 8 and Table 1) [26].

Statistical analysis involved calculating medians (p50) and quartiles (p25 and p75), establishing median-based regression models for age and sex relationships, and Pearson coefficients for inter-side and variable relationships with height, weight, and BMI. A median-based calculation was used to minimize the effects of confounders. Reporting non-parametric measures should ensure consistency across all variables concerning data distribution and presentation [27]. We used the Pearson coefficient for the relationship between the right and left clavicles because they are simple linear dependencies. For all the other values, a beta coefficient was calculated because these analyses took place in independent, multiple-regression models. We considered minor impact for a Pearson coefficient ≥ 0.1, moderate impact for beta ≥ 0.3, and major impact for beta ≥ 0.5, admitting that these thresholds are context-dependent and may not be universally applicable [28]. A significance threshold of *p* < 0.05 was applied. Stata 17.0 (Stata Corporation, College Station, TX, USA) facilitated statistical computations. 

### 2.4. Ethics

Ethical approval (BB 39/08, 19 June 2008) and individual consent were obtained, aligning with the SHIP study protocols. This study, registered as a SHIP project with the ID SHIP/2019/01/D, operates within the Community Medicine Institute of the University of Greifswald and is financially supported by various government entities. 

## 3. Results

In the course of this study, MRIs from a final cohort of n = 2436 participants were meticulously examined. However, two participants were excluded due to duplicated labeling, and an additional n = 879 images were deemed of low quality (Figure 9).

For the construction of the reference table, 845 participants out of the initial 2436 were excluded based on specific criteria (Figure 10 and Table 2): edema (539), effusion (559), perimeter extensions (13), glenoid retroversion (159), nearby fracture (25), or pain intensity in the shoulder >3 (301). The purpose was to exclude as many pathological/abnormal SCJs and clavicles from the normative table as possible in order to represent a shoulder-healthy population. Some participants exhibited more than one of these features.

The median (P50) of the medial maximal diameter and the lateral maximal diameter are comparable, indicating uniform thickness on both ends. The largest diameter measures 30.67 mm, while the thinnest is 3.25 mm.

There is a significant correlation between diameters on the right and left sides (Figure 11): For Mincl with 0.77, Maxlcl with 0.71, Maxmcl with 0.7, and Maxcl with 0.67. This suggests a tendency for symmetry in clavicle diameters.

Older patients do not seem to have a tendency for thinner or thicker clavicles (Table 3). 

Female participants exhibit smaller diameters compared to their male counterparts (Table 4). Notably, this indicates beta values, e.g., Maxlcl on the right side (beta = −2.89) and left side (beta = −2.65), as well as Mincl on the right side (beta = −2.23) and left side (beta = −1.99). Diameters are in median values thinner in females.

Height, weight, and BMI exert less influence on the variables compared to gender (Table 5). 

The height has a minor impact on the left Ca (beta = −0.10) and Csb (beta = −0.14) and a small influence on the right Maxlcl (beta = 0.14) and the left Maxlcl (beta = 0.14).

Diameters show less sensitivity to weight, illustrated by the decrease in beta values for Maxlcl from 0.14 (height) to 0.06 (weight) on the right side and from 0.14 (height) to 0.05 (weight) on the left side.

Overall, weight does not have a beta > 0.1, suggesting minimal relevance to the variables. BMI, with beta values of 0.2 and 0.21, primarily impacts Sa_r and Sa_l.

## 4. Discussion

The clavicle, a frequently fractured bone, and its adjacent SCJ are critical subjects for research aimed at enhancing our understanding and potential for restoration [29]. Despite the importance of these factors and the increasing prevalence of MRI, investigations into MR imaging of the clavicle and the SCJ remain limited. Nevertheless, some associations have been made, revealing correlations between clavicular parameters, sex, and age, primarily through CT scans and cadaver examinations of removed and prepared clavicles [12,30]. Existing literature also explores the measured asymmetry of the right and left clavicles in cadavers via direct measurement from removed, washed, and boiled clavicles as well as from radiographs [12,31,32]. In our reference table, the maximal clavicular diameter differs only by one millimetre. In addition to the site dependency from right/left clavicle diameters, it enables thoughts about planning plate osteosynthesis for clavicle fractures with more precise preparation, which seems to lead to better outcomes [33,34]. 

Several studies have elucidated the relationship between clavicular parameters and the age and sex of patients [13,30,35,36,37]. It is evident that gender plays a more significant role, particularly in the parameters examined in this study. Prior research employing the regression coefficient demonstrated a small correlation between height and diameter [16]. Our study, with a larger sample size, yielded comparable results, indicating less impact with beta values of 0.08 for the right, 0.07 for the left medial diameter, and 0.14 for both the right and left lateral diameter. The increased precision of larger studies facilitates clearer distinctions and minimizes the likelihood of error [38]. Moreover, variations in technical approaches and precision among diagnostic tools, such as X-rays, CT scans, MRIs, and cadaver examinations, contribute to discrepancies in results [11,39,40].

While the height of an individual appears to influence the diametric shape of the clavicle, further research is warranted to discern the extent of its impact. For the differentiation between men and women, substantial differences were detected, with beta values ranging from −1.76 to −2.93, emphasizing the significant dissimilarity in skeletal structure between genders and the tendency for smaller diameters in female patients [12,30]. Interestingly, there is a dearth of literature investigating the relationship between parameters and BMI, or weight. Only the risk of fractures has been evaluated, but it seems to be inconclusive [41]. Our study suggests that weight is a less critical factor than BMI, possibly due to the higher beta associated with height contributing to BMI and potentially elevating the correlation value of weight. However, we acknowledge that thresholds indicating the strength of the effect sizes concerning correlation coefficients are always context-dependent, somehow subjective depending on the interpreter, and may not be universally applicable. 

### Comparative Analysis

Our results were compared with other research data (Table 6). The number of analyzed images varies for each variable. It is noteworthy that studies with smaller patient cohorts show greater deviations from our results than those with larger sample sizes. For example, Bernat’s study [14] exhibits a difference of approximately 3 mm in maximal lateral diameter (LD) in males and about 4 mm in LD right, compared to Jagmahender’s study [12], which has only a 1 mm difference from our values in LD males. Our values for maximal lateral, maximal medial, and minimal diameter are consistently one to four millimeters lower than those reported in other studies [12,14,15,16,42]. This could be attributed to the fact that two articles used cadavers; one used CT scans, while we used MRI. Additionally, the number of patients in our study is significantly higher than in the other studies. It should be noted that we utilized a median-based description, while other researchers employed an average-based presentation. Despite this difference, with a large number of patients, the median should not deviate significantly from the average value.

The interpretation of results is also influenced by the investigation of different populations with varying average heights, weights, and geographical variants [12,14,15,16,42]. For example, our study analyzed people from Mecklenburg-Vorpommern, who are predominantly considered to be Caucasian, while other research focused on individuals from Asian countries [12,16]. Anthropological variations may contribute to divergent results. Methodological limitations include variations in MRI imaging and anatomical differences, along with the inherent inaccuracies in imaging bony structures. MRI is particularly well suited for soft tissue visualization, which may contribute to inaccuracies in the assessment of bony anatomy. While CT remains superior for detailed bony evaluations, MRI is increasingly being used for shoulder assessments, including the acromioclavicular and sternoclavicular joints. Although exclusion criteria with a probable association with shoulder girdle pathologies were applied to reduce confounders, it is possible that not all influencing factors were completely excluded, such as undetected underlying medical conditions. The location of the lateral clavicular end, which is more cranially positioned than the medial end, may lead to its absence in some cases. As a result, more medial diameters were scanned in the transversal layer and could be evaluated. Additionally, angles were infrequently analyzed, limiting comparisons with other scientific work. While our MRI diagnostic was standardized with an MRI imaging protocol, slight patient movement during the examination cannot be entirely excluded [24].

In conclusion, despite these limitations, our study provides valuable reference values for clavicle anatomy and the corresponding medial joint. These findings contribute to an improved understanding of healthy shoulder girdle anatomy. Further investigations are imperative to examine the clinical implications of these reference values in patients with sternoclavicular and clavicular pathologies, establishing a foundation for future research on this topic. Incorporating clinical outcome data in future studies is essential for validating the clinical relevance of these normative values in fracture patterns and surgical decision-making.

## 5. Conclusions

In summary, our study delved into the relatively underexplored realm of magnetic resonance imaging (MRI) of the clavicle and its adjacent sternoclavicular joint (SCJ), aiming to enrich our comprehension and potential for restoration of these critical anatomical structures. Through the analysis of a substantial cohort of 2436 participants, we observed notable associations between clavicular parameters, sex, and age, shedding light on the intricate interplay of these variables. Our findings underscore the substantial influence of gender on clavicular morphology, with men exhibiting significantly different skeletal structures compared to women. Moreover, we explored the impact of height, weight, and body mass index (BMI) on clavicular dimensions, revealing intriguing insights into the lesser role of weight compared to BMI. This suggests a complex interplay between anthropometric variables and clavicular anatomy, warranting further investigation. Our study, with its robust sample size and standardized MRI imaging protocol, offers valuable reference values for clavicle anatomy for Caucasian people, enhancing our understanding of shoulder girdle morphology. Despite certain methodological limitations, such as variations in MRI imaging and anatomical considerations, our study lays a solid foundation for future research in this domain. The comprehensive dataset generated provides a basis for exploring the clinical implications of our findings in patients with sternoclavicular and clavicular pathologies, ultimately contributing to advancements in orthopedic medicine and patient care.

## Figures and Tables

**Figure 1 jcm-13-03598-f001:**
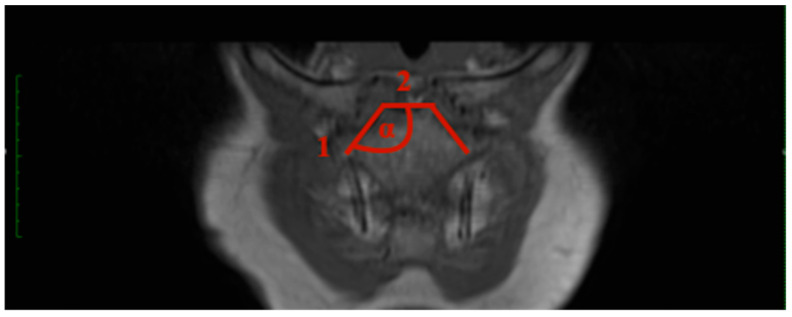
How to measure the sternal angle (SA): Angle between the articular surfaces of the sternoclavicular joint and the sternum, utilizing tangents drawn from specific points, T1 weighting the 3D frontal layer with a slice thickness of 3 mm.

**Figure 2 jcm-13-03598-f002:**
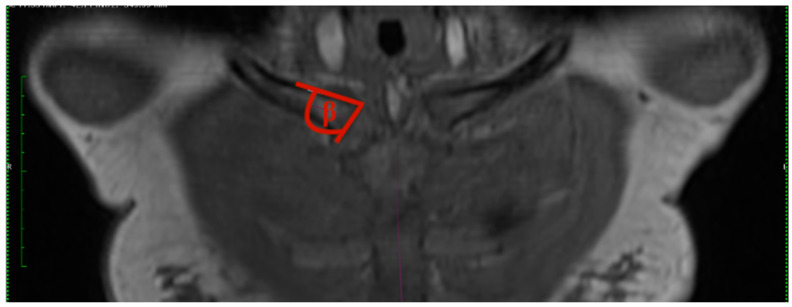
How to measure clavicular angle (CA): angle representing the inclination between the articular surface of the SCJ and the clavicle, derived from tangents and forming the acute angle beta, T1 weighting the 3D frontal layer with a slice thickness of 3 mm.

**Figure 3 jcm-13-03598-f003:**
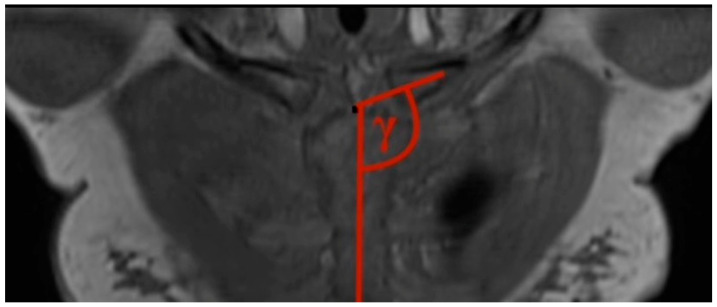
How to measure the clavicular—sternal angle (CSB): The vertical axis of the sternum, deriving from a line drawn from the middle point of the cranial end of the manubrium to the xiphoid process, and the lengthwise axis of the clavicle, drawn from the middle of the lateral clavicle to the middle of the medial clavicle, produced the obtuse angle gamma. T1 weights the 3D frontal layer with a slice thickness of 3 mm.

**Figure 4 jcm-13-03598-f004:**
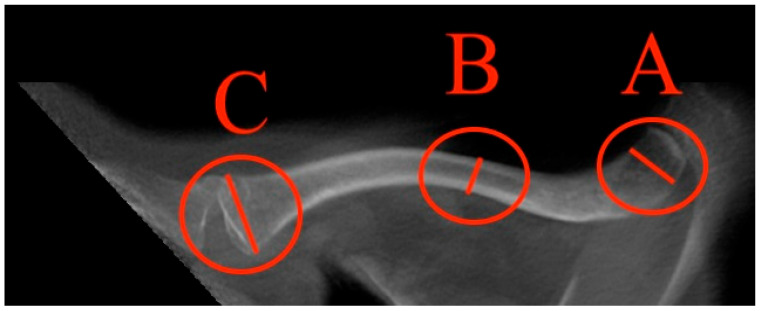
Schematic figure of the superior view of the clavicle bone illustrates the measurement areas and examples of measurement for A = maximal lateral diameter, B = minimal diameter, and C = maximal medial diameter.

**Figure 5 jcm-13-03598-f005:**
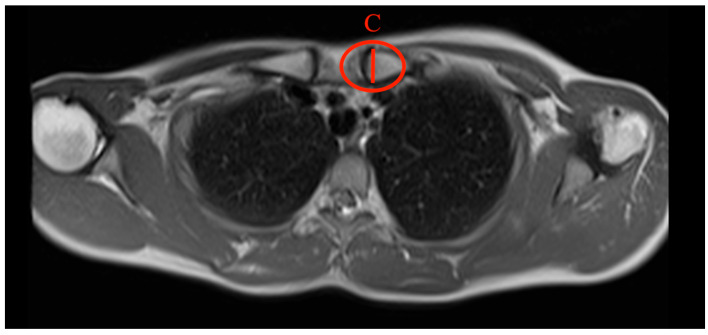
How to measure the max. medial diameter (MAXMCL): Measured at the medial end of the clavicle from the furthest ventral to dorsal points with an orthogonal line to the clavicle axis, T1 weighting has a slice thickness of 3 mm (C).

**Figure 6 jcm-13-03598-f006:**
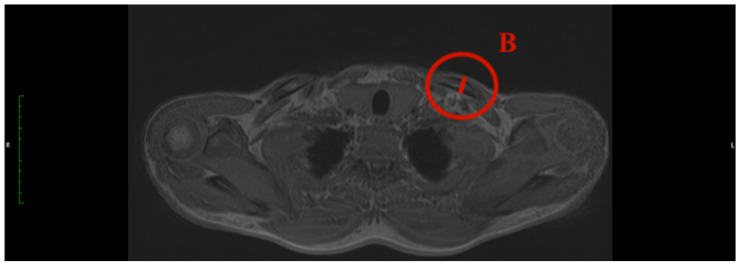
How to measure the min. diameter (MINCL): Measured similarly to MAXMCL but focusing on the least distant ventral to dorsal points of the medial clavicle, T1 weighting has a slice thickness of 3 mm (B).

**Figure 7 jcm-13-03598-f007:**
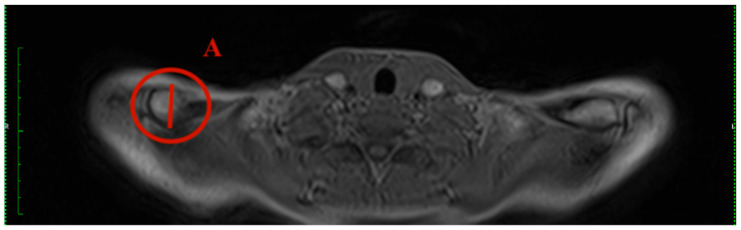
How to measure the max. lateral diameter (MAXLCL): Measured at the lateral end of the clavicle, from the furthest ventral to dorsal points, with an orthogonal line to the clavicle axis, T1 weighting has a slice thickness of 3 mm (A).

**Figure 8 jcm-13-03598-f008:**
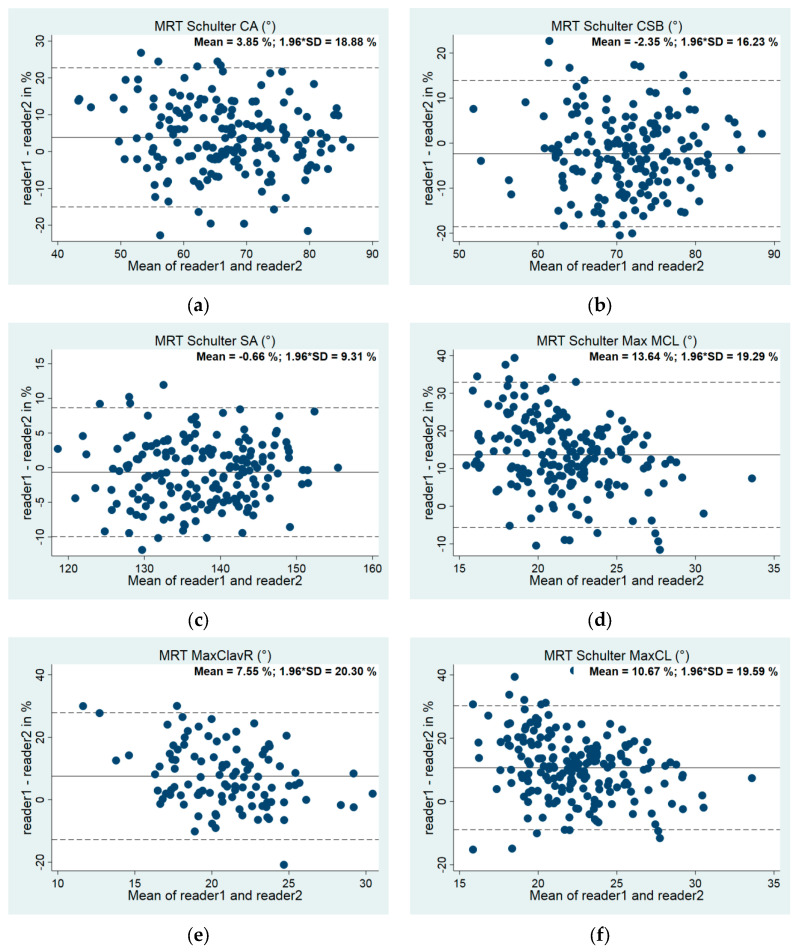
(**a**–**g**) Bland–Altman plots for interreader variability: (**a**) Clavicular angle (CA, n = 177), (**b**) clavicular–sternal angle (CSB, n = 174), (**c**) sternal angle (SA, n = 176), (**d**) max. medial diameter (Max MCL, n = 189), (**e**) max. lateral diameter (MaxClavR, n = 104), (**f**) max. clavicular diameter (MaxCL, n = 196), (**g**) minimal clavicular diameter (MinCL, n = 174).

**Figure 9 jcm-13-03598-f009:**
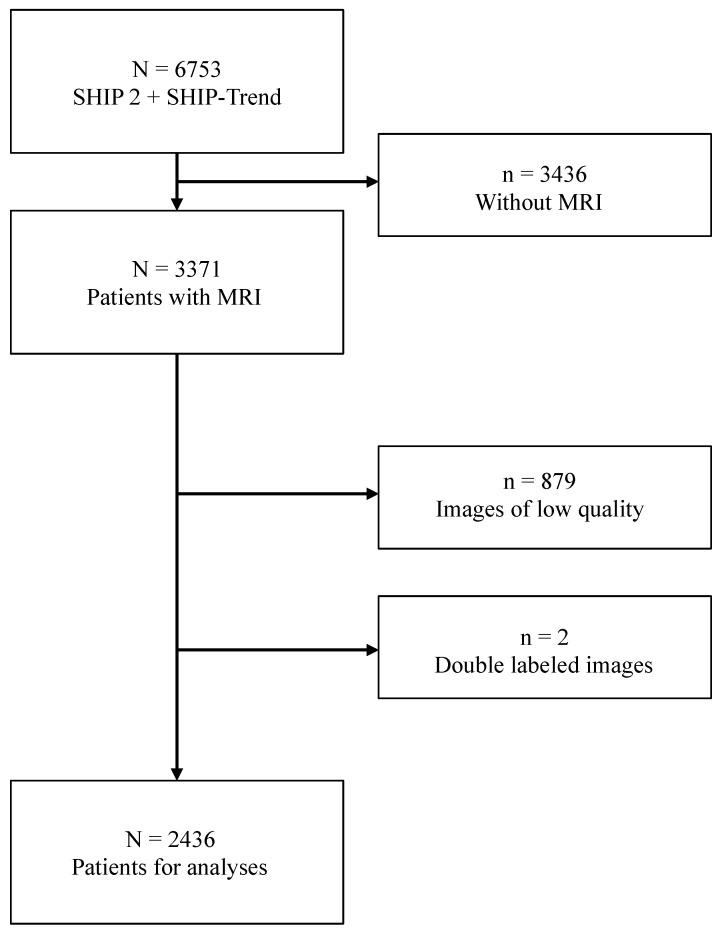
Selection of the participants (composition of analyzed MRI scans and number of rolled-out patients).

**Figure 10 jcm-13-03598-f010:**
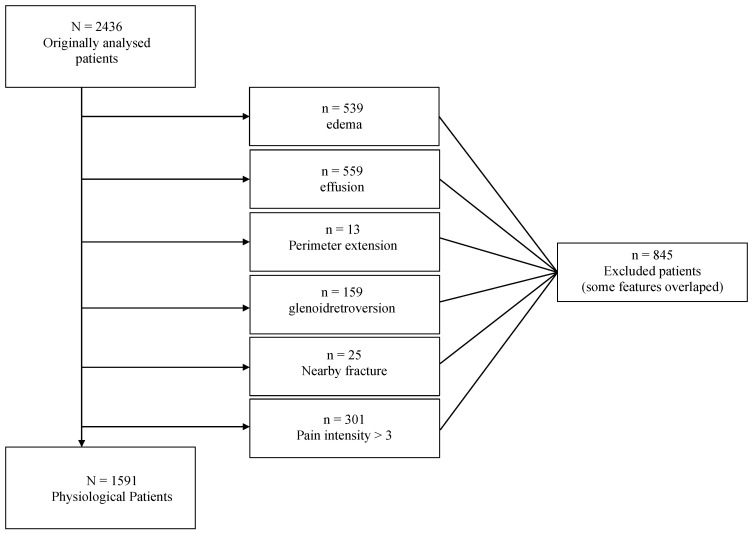
Excluded patients for the reference table due to the exclusion criteria in order to define a shoulder-healthy population.

**Figure 11 jcm-13-03598-f011:**
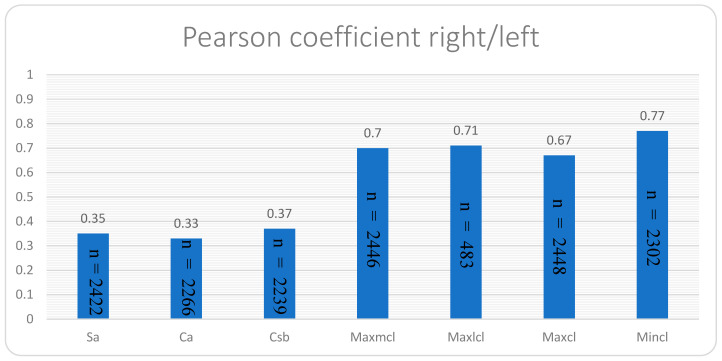
Showing the Pearson coefficient for all measured variables by comparing the right and left sides.

**Table 1 jcm-13-03598-t001:** Interrater variability. Amount of measurements (n), Interclass correlation coefficients (ICC), mean bias (absolute, in % to 360° and their standard deviations (SDs)) [26].

Variable	n			ICC	95% Konf. Interval	Mean Value		Inter Absolut		Inter %	
	R1	R2	Both			R1	R2	Mean	SD	Mean	SD
med_sa	182	182	177	<0.001	na	137.00	137.89	−0.89	6.47	−0.66	4.75
med_ca	180	178	174	0.013	0.001; 0.178	67.83	65.39	2.44	6.28	3.85	9.63
med_csb	176	181	176	0.007	0.000; 0.200	70.64	72.34	−1.70	5.81	−2.35	8.28
med_maxmcl	189	198	189	0.154	0.024; 0.579	23.03	20.19	2.84	2.01	13.64	9.84
MaxClavRlateral	104	105	104	0.034	0.003; 0.300	21.61	20.14	1.47	2.09	7.55	10.36
med_maxcl	196	198	196	0.105	0.015; 0.478	23.46	21.17	2.29	2.09	10.67	10.00
med_mincl	174	177	174	<0.001	na	7.59	7.44	0.16	1.10	2.99	15.45

**Table 2 jcm-13-03598-t002:** Reference table with values deriving only from shoulder-healthy patients after application of the exclusion criteria (angles in degree and diameter in mm).

Variable	N	Male	Female	P50	P25	P75	Min.	Max.
Sa_r	1249	610	639	133.77	127.71	140.89	113.72	171.21
Sa_l	1249	610	639	132.95	126.79	138.77	102.19	159.29
Ca_r	1181	572	609	61.99	54.24	70.72	35.41	89.9
Ca_l	1199	580	619	68.53	60.39	76.3	35.03	89.85
Csb_r	1172	570	602	64.66	57.73	70.69	42.15	88.5
Csb_l	1185	576	609	69.46	63.42	74.71	45.19	88.53
Maxmcl_r	1261	612	649	20.84	18.56	22.98	13.32	30.16
Maxmcl_l	1262	614	648	20	18.01	22.22	12.42	30.56
Maxlcl_r	410	150	260	20.86	18.46	23.32	12.05	29.95
Maxlcl_l	283	90	193	20.19	17.98	22.84	10.38	30.67
Maxcl_r	1261	612	649	21.46	19.31	23.55	13.32	30.16
Maxcl_l	1263	614	649	20.49	18.45	22.76	12.42	30.67
Mincl_r	1206	576	630	8.37	6.95	9.74	3.25	15.51
Mincl_l	1195	570	625	8.05	6.75	9.39	3.52	13.70

**Table 3 jcm-13-03598-t003:** Influence of age (showing a coefficient for a median-based logistic regression).

Parameter	Right Side		Left Side	
	N	β for Age (95%—CI)	N	β for Age (95%—CI)
SA	2420	−0.07 (−0.10; −0.04)	2420	−0.06 (−0.09; −0.03)
CA	2275	−0.2 (−0.24; −0.15)	2308	−0.21 (−0.25; −0.17)
CSB	2250	−0.00 (−0.04; 0.03)	2274	0.03 (−0.00; 0.06)
MaxMCL	2446	0.02 (0.01; 0.03)	2447	0.01 (0.00; 0.02)
MaxLCL	753	0.01 (−0.01; 0.03)	512	0.01 (−0.01; 0.04)
MaxCL	2447	0.02 (0.01; 0.03)	2449	0.01 (0.00; 0.02)
MinCL	2325	−0.01 (−0.02; −0.00)	2303	−0.01 (−0.01; −0.00)

**Table 4 jcm-13-03598-t004:** Influence of sex (showing a coefficient for a median-based logistic regression).

Parameter	Right Side		Left Side	
	N	β for Women vs. Men (95%—CI)	N	β for Women vs. Men (95%—CI)
SA	2420	−0.26 (−1.18; 0.66)	2420	0.82 (−0.02; 1.67)
CA	2275	−2.28 (−3.49; −1.08)	2308	−1.18 (−2.36; −0.01)
CSB	2250	−1.76 (−2.79; −0.72)	2274	0.28 (−0.49; 1.06)
MaxMCL	2446	−2.93 (−3.22; −2.63)	2447	−2.91 (−3.19; −2.64)
MaxLCL	753	−2.89 (−3.53; −2.25)	512	−2.65 (−3.44; −1.86)
MaxCL	2447	−2.60 (−2.88; −2.32)	2449	−2.58 (−2.88; −2.29)
MinCL	2325	−2.23 (−2.42; −2.04)	2303	−1.99 (−2.16; −1.82)

**Table 5 jcm-13-03598-t005:** Influence of height, weight, and BMI (showing a coefficient for a median-based regression model).

Variables	Height Coefficient	Weight Coefficient	BMI Coefficient
Sa r	−0.02	**0.05**	**0.2** *
Sa l	−0.001	**0.05**	**0.21** *
Ca r	0.08	0.01	−0.04
Ca l	**−0.10** *	**−0.05**	−0.09
Csb r	−0.05	−0.01	−0.01
Csb l	**−0.14** *	−0.02	0.01
Maxmcl r	**0.08**	**0.05**	**0.09** *
Maxmcl l	**0.07**	**0.04**	**0.08** *
Maxlcl r	**0.14** *	0.06	0.11
Maxlcl l	**0.14** *	**0.05**	**0.08**
Maxcl r	**0.07** *	**0.03**	**0.04**
Maxcl l	**0.07** *	**0.03**	**0.06**
Mincl r	**0.03**	**0.04**	**0.11** *
Mincl l	**0.04**	**0.04**	**0.08** *

Results are derived from median regression models; CI = confidence interval; bold printed = *p* < 0.05; * = greatest impact/correlation for the variable.

**Table 6 jcm-13-03598-t006:** Comparing our results with other data derived from different diagnostic methods, as seen below, all parameters are presented in millimeters, and except for our values, they are all mean values. (MD = maximal medial diameter, LD = maximal lateral diameter, MinD = minimal diameter, r = right, l = left).

Parameters	Our Research N = 1591	Our Q1–Q3	Jesse Chieh Szu Yang [16] N = 100	Andermahr [15] N = 196	Duprey [42] N = 6	Jagmahender Singh Sehrawat [12] N = 263	Amit Bernat [14] N = 34
Diagnostic tool	MRI	MRI	CT	Cadaver	X-Ray	Cadaver	Cadaver
MD male	22.21 r 21.52 l	20.34–24.47 r 19.4–23.5 l	25 ± 3,	26 ± 4	-	22.09 ± 3.30 r 21.32 ± 2.93 l	24.7 ± 2.8
MD female	19.64 r 18.73 l	17.53–21.27 r 16.89–29.66 l	23 ± 3	24 ± 4	-	18.82 ± 2.72 r 18.12 ± 2.63 l	22.8 ± 2.8
MD right	20.84	18.56–22.98	-	25 ± 4	-	-	24.3 ± 3
MD left	20	18–22.22	-	25 ± 4	-	-	23.2 ± 2.8
MD total	-		24 ± 3	25 ± 4	26 ± 3	-	23.8 ± 3
LD male	22.82 r 22.88 l		26 ± 4	24 ± 4	-	23.5 ± 3.46 r 23.76 ± 3.50 l	25.9 ± 4.1
LD female	19.94 r 19.39 l		22 ± 3	21 ± 4	-	20.71 ± 3.01 r 20.82 ± 3.26 l	23.5 ± 3
LD right	20.86	18.46–23.32	-	23 ± 4	-	-	24.8 ± 3.7
LD left	20.19	17.98–22.84	-	23 ± 4	-	-	24.7 ± 3.9
LD total	-		24 ± 4	22 ± 4	27 ± 4	-	24.7 ± 3.8
MinD right	8.37	6.95–9.74	-	-	-	-	10.9 ± 1.6
MinD left	8.05	6.75–9.39	-	-	-	-	10.8 ± 1.5

## Data Availability

The data presented in this study are available within the manuscript.

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
