# Peer review of "Normative Values for Sternoclavicular Joint and Clavicle Anatomy Based on MR Imaging: A Comprehensive Analysis of 1591 Healthy Participants"

_jcm, 2024, doi:10.3390/jcm13123598_

Round 1

Reviewer 1 Report

Comments and Suggestions for Authors

Dear authors,

Thank you for the opportunity to review the manuscript entitled:  “Normative Values for Sternoclavicular joint and clavicle anatomy based on MR-Imaging: A Comprehensive Analysis of 2436 Healthy Participants.“

            The manuscript is easy to read, the scientific information being presented in a logical sequence. The study stands out due to the large number of included subjects and their selection method. However, the proposed topic does not bring novelties from a scientific point of view, as numerous morphometric studies of the clavicle have been carried out. The major difference between these and the present study is the imaging modality through which it takes place. But MRI compared to CT has two major disadvantages: the duration of the exploration and the high cost. Precisely these disadvantages do not recommend MRI exploration in the case of medical emergencies such as fractures. The majority of orthopaedists use CT scanning in the operative planning in the case of clavicle fractures. Because the MRI provides important data regarding the articular components, a reconversion of the study and the assessment of the sterno-clavicular and acromio-clavicular joints before and after the intervention would be useful.

In addition, I identified several aspects that can improve the quality of this article:

*         The abstract must be organized in the format characteristic of an original article

*         Lines 30-31 and 33-34 - the information is repeated

*         Figure 1 - the image does not reflect the content. The beta (clavicular) angle is shown.

*         Figure 4 - The line used to measure the lateral extremity of the clavicle does not respect the orthogonal line to the clavicle axis criterion. Consequently, if the measurements were not performed according to the orthogonal criterion, the results must be revised.

*         Lines 277 and 279  - for an easier reading, I recommend positioning the reference immediately after the authors' names: Bernat's et al.[13]

Best regards.

Reviewer 2 Report

Comments and Suggestions for Authors

This is a well-designed and comprehensive study that establishes normative values for clavicle and sternoclavicular joint anatomy based on a large sample of MRI scans from a healthy population.

There are a few minor issues:

1.     Lack of information on the racial/ethnic composition of the study population, or just Caucasians, which could limit the generalizability of the findings, please mention this in the limitation. 

2.     No details provided on the exclusion criteria used to define the "shoulder-healthy" population, which could introduce some selection bias.

3.     Potential for measurement errors or inaccuracies inherent to MRI-based assessments of bony anatomy.

4.     Lack of clinical outcome data to directly link the normative values to relevant clinical implications, such as fracture patterns or surgical decision-making.

5.     The introduction could be more concise and focused, as it includes some unnecessary details (e.g., the repetition of the clavicle fracture statistic).

6.     The background does not provide a clear and comprehensive overview of the existing knowledge on clavicular anatomy and the use of MRI versus x-ray in its assessment. Some additional background information on these topics would strengthen the introduction.

7.     The introduction could better highlight the clinical relevance and potential impact of the study, beyond just the need for informed decision-making in clavicle fracture treatment.

8.     The introduction could be strengthened by more clearly articulating the specific epidemiological hypotheses the study aims to incorporate, as this is an important aspect of the research.

9.     Exclusion Criteria:

The exclusion criteria for the reference table, such as edema, effusion, and shoulder pain intensity, are clearly stated, but it would be helpful to know the rationale behind these specific criteria.

10.  Sample Size:

The sample size of 2,436 individuals for the final analysis is not explicitly stated in the provided text, but can be inferred. It would be useful to have this information clearly reported.

11.  Statistical Analysis:

The details of the statistical analysis methods used to derive the results and conclusions are not provided in this excerpt. Including this information would strengthen the transparency and replicability of the study.

12.  Small sample size for some variables: The sample size for certain variables (e.g., MaxClavRlateral, n = 104) is relatively small, which may limit the reliability and generalizability of the findings.

13.  Interpretation of effect sizes: The authors have provided thresholds for interpreting the magnitude of Pearson and beta coefficients (small, moderate, and large), but these thresholds are somewhat subjective and may not be universally applicable.

14.  Lack of information on data distribution: The authors have not provided any information on the distribution of the data (e.g., normality tests), which could have informed the choice of statistical tests.

15.  Potential confounding factors: While the authors have examined relationships with anthropometric variables, they have not addressed the potential influence of other factors, such as age, sex, or underlying medical conditions, on the measured clavicular parameters.

Overall, this is a high-quality study that makes an important contribution to the literature by providing much-needed reference values for clavicle and sternoclavicular joint anatomy. The findings have the potential to enhance clinical practice, particularly in the context of clavicle fracture management and surgical planning. Further research linking these normative values to clinical outcomes would be a valuable next step.

Round 2

Reviewer 1 Report

Comments and Suggestions for Authors

Dear authors,

Thank you for the opportunity to re-review the manuscript entitled “Normative Values for Sternoclavicular joint and clavicle anatomy based on MR-Imaging: A Comprehensive Analysis of 2436 Healthy Participants. “

The authors took into account the suggestions offered and made the indicated changes. The quality of the manuscript has improved in terms of the presentation of the study. I am referring first to the Materials and Methods chapter in which the numerical data of patient selection as well as the images were corrected. Because of the new selection criteria applied, I recommend changing the title because the study itself is conducted (according to the exclusion criteria) on 1591 patients.

Although from a clinical point of view I consider that the present study does not offer significant implications, it represents a first step in establishing an anatomical norm within MRI investigations.  

Best regards.
